# Oncomine™ Comprehensive Assay v3 vs. Oncomine™ Comprehensive Assay Plus

**DOI:** 10.3390/cancers13205230

**Published:** 2021-10-18

**Authors:** Lau K. Vestergaard, Douglas N. P. Oliveira, Tim S. Poulsen, Claus K. Høgdall, Estrid V. Høgdall

**Affiliations:** 1Molecular Unit, Department of Pathology, Herlev Hospital, University of Copenhagen, DK-2730 Herlev, Denmark; lau.kraesing.vestergaard@regionh.dk (L.K.V.); douglas.nogueira.perez.de.oliveira@regionh.dk (D.N.P.O.); tim.svenstrup.poulsen@regionh.dk (T.S.P.); 2Juliane Marie Centre, Department of Gynecology, Rigshospitalet, University of Copenhagen, DK-2100 Copenhagen, Denmark; claus.hogdall@regionh.dk

**Keywords:** targeted NGS, genomic profiling, biomarker discovery, clinical research, Oncomine Comprehensive Assay v3, Oncomine Comprehensive Assay Plus

## Abstract

**Simple Summary:**

The detection of genetic alterations in cancer is important to obtain knowledge of the underlying mutational tumor composition. Knowing the mutational profile can assist oncologists on tailoring optimal personalized treatments. Moreover, obtaining additional information from a broader cancer-related gene panel, without compromising performance, can benefit both current and future patients. In this study, we assessed the performance of gene mutations identified from sequencing using the newly Oncomine™ Comprehensive Assay Plus (OCA-Plus). The assessment was performed in comparison to gene mutations identified from sequencing using the Oncomine™ Comprehensive Assay v3 (OCAv3), currently used in our routine clinical setting. Therefore, an investigation of their performance was conducted on intersecting nucleotide positions within overlapping genes covered by both the OCA-Plus and the OCAv3. We show here that there is a 91% concordance between identified pathogenic and likely pathogenic classified variants.

**Abstract:**

The usage of next generation sequencing in combination with targeted gene panels has enforced a better understanding of tumor compositions. The identification of key genomic biomarkers underlying a disease are crucial for diagnosis, prognosis, treatment and therapeutic responses. The Oncomine™ Comprehensive Assay v3 (OCAv3) covers 161 cancer-associated genes and is routinely employed to support clinical decision making for a therapeutic course. An improved version, Oncomine™ Comprehensive Assay Plus (OCA-Plus), has been recently developed, covering 501 genes (144 overlapping with OCAv3) in addition to microsatellite instability (MSI) and tumor mutational burden (TMB) assays in one workflow. The validation of MSI and TMB was not addressed in the present study. However, the implementation of new assays must be validated and confirmed across multiple samples before it can be introduced into a clinical setting. Here, we report the comparison of DNA sequencing results from 50 ovarian cancer formalin-fixed, paraffin-embedded samples subjected to OCAv3 and OCA-Plus. A validation assessment of gene mutations identified using OCA-Plus was performed on the 144 overlapping genes and 313,769 intersecting nucleotide positions of the OCAv3 and the OCA-Plus. Our results showed a 91% concordance within variants classified as likely-pathogenic or pathogenic. Moreover, results showed that a region of *PTEN* is poorly covered by the OCA-Plus assay, hence, we implemented rescue filters for those variants. In conclusion, the OCA-Plus can reflect the mutational profile of genomic variants compared with OCAv3 of 144 overlapping genes, without compromising performance.

## 1. Introduction

Next generation sequencing (NGS) is increasingly being employed in clinical oncology for the genomic profiling of tumor samples. Targeted exome sequencing (TES), including the sequencing of hot-spot regions, aims at the identification of genetic variants in specific genes, which are known genomic biomarkers important for diagnosis, prognosis, treatment and therapeutic response [1,2]. The Oncomine™ Comprehensive Assay v3 (OCAv3) covers 161 cancer-associated genes, allowing the detection of single nucleotide variants (SNV), multiple-nucleotide variants (MNV) and small insertions/deletions (indel). The OCAv3 has, since December 2017, been routinely implemented in our clinical setting to assist oncologists’ decisions on therapeutic courses. The performance of OCAv3 was recently used to focus treatment options for refractory metastatic colorectal cancer [3]. 

Ovarian and breast cancer patients with pathogenic mutations in the *BRCA 1/2* genes have shown to benefit from therapeutic intervention with poly (ADP-ribose) polymerase inhibitor drugs [4,5]. A list of approved drugs for the treatment of ovarian and breast cancer is outlined in Table 1. Ovarian cancer (OC) is one of the most lethal types among gynecological malignancies, with up to 70% of the incidences harboring advanced tumor stages (Federation of Gynecology and Obstetrics (FIGO III-IV)) [6,7]. OC is categorized into four groups based on the histological subtypes: serous-, mucinous-, endometroid- and clear cell adenocarcinoma. Moreover, those tumors can be further classified as low- or high-grade.

OC harbors a heterogeneous molecular genotype with diverse pathologic characteristics. Thus, the underlying genomic landscape of OC is suited for evaluating performance of new assays. For instance, high-grade serous OC shows a mutational prevalence of the *TP53* gene [8,9]. Moreover, somatic mutations and deleterious somatic mutations are frequently observed in genes associated with homologous recombination repair (HRR). 

The Oncomine™ Comprehensive assay Plus (OCA-Plus) covers 501 cancer-associated genes, of these, 144 are overlapping with OCAv3. From the 144 genes, the 2 panels harbor 313,769 intersecting nucleotide positions from which performance assessment of variants will be validated on in this study. OCA-Plus also include microsatellite instability (MSI) and tumor mutational burden (TMB) assays, providing a time-efficient single workflow. The addition of MSI and TMB may provide information of potential immunotherapies for patients with solid tumors [10]. However, high-grade serous adenocarcinoma is often not harboring a high number of mutations (TMB-high) reflected in low genomic instability. Therefore, samples included in this comparison study are not optimal for the validation of MSI and TMB. Thus, MSI and TMB are not validated by the presented study.

The OCA-Plus is covering the 15 HRR genes listed in the Lynparza HHR gene panel by their full length, whereas OCAv3 covers a subset of these genes (Appendix A). Hence, OCA-Plus might contribute to select additional candidates for Olaparib treatment and additionally might provide a beneficial perspective of understanding HRR deficient cancers better, thus improving personalized treatment schemes. Genes associated with the PI3K pathway (phosphoinositide 3-kinase (PI3K)/AKT/mechanistic target of rapamycin (mTOR)) are found to be mutated in 70% of OC [11] and, therefore, an important pathway for personalized treatment options via inhibition of PI3K-signalling. Therefore, identifications of deleterious mutations in *PIK3CA* and *PTEN* are of clinical relevance for treatment tailoring.

Formalin-fixed and paraffin-embedded (FFPE) tissue is commonly used in clinical routine settings due to its versatility of molecular characterizations of tumors. It is applicable to a broad range of molecular techniques, such as immunohistochemistry and in situ hybridization assays.

However, formalin causes the deamination of cytosine, producing a base substitution of C to T or G to A (on the antisense strand), hence generating irreversible false-positive sequencing artifactual variants that may compromise interpretation of low-frequency variants. Being that C:G > T:A transitions are the predominant signatures associated with FFPE samples, some reports have shown that treatment with uracil DNA glycosylase (UDG) reduce these deamination artifacts significantly [12,13,14]. Because these artifactual mutations happen with low allelic frequency and additionally occur in tumor-free samples, these can be classified as artifactual changes, thus: false positives changes [15].

The larger gene-panel in OCA-Plus may potentially provide novel insights into (1) the stratification of patients, (2) prioritizing genes for future studies and (3) the development of novel molecular drugs. 

In this study, we investigated and evaluated the concordance between variants identified using OCAv3 and OCA-Plus in 50 ovarian cancer FFPE samples. We additionally set quality parameters and thresholds for filtering variants and restoring potential true variants. Moreover, we reported observations with the OCA-Plus in specific *loci* areas that need caution when interpreting variants. 

## 2. Materials and Methods

### 2.1. Patients and Samples Handling

A total of 50 tissue samples were retrospectively collected from a prospective cohort of OC patients [16,17]. Patients were registered in the Pelvic Mass study/GOVEC study, a prospective cohort initiated in September 2004. Clinical information of the patients is registered in the Danish Gynecological Cancer Database (DGCD) [18].

Surgery was performed at the Department of Gynecology, Rigshospitalet, and all tissue was handled at the Department of Pathology, Rigshospitalet. All samples were registered and stored in the Danish Cancer Biobank (DCB, Bio- and GenomeBank, Denmark—http://rbgb.dk/cancer/) (accessed on 16 August 2021) under their defined conditions according to national biobank guidelines. The cohort was examined and classified by expert gynecologists, as follows: 39 high grade serous adenocarcinoma, 4 clear cell adenocarcinoma, 4 mucinous adenocarcinoma and 3 endometrioid adenocarcinoma. 

### 2.2. DNA Extraction

Area of cancerous tissue in FFPE blocks were identified and subsequently extracted by 1-mm disposable punchers in order to assure high content of tumor cells. Genomic DNA was extracted using Maxwell^®^ RSC DNA FFPE Kit (Promega, Madison, WI, USA) following manufacturer’s instructions. 

DNA concentration was quantified using Qubit™ ds DNA High-Sensitive Assay kit (Thermo Fisher Scientific, Waltham, MA, USA) on the Qubit fluorometer (Thermo Fisher Scientific, Waltham, MA, USA).

### 2.3. Library Preparation and Sequencing 

All library preparation was performed manually for OCAv3 (Thermo Fisher Scientific, Waltham, MA, USA) and OCA-Plus (Thermo Fisher Scientific, Waltham, MA, USA) according to manufacturer’s instructions MAN0015885 (Revision C.0) and MAN0018490 (Revision D.0), respectively. Multiplex PCR amplification was conducted using a DNA concentration of approximately 20 ng as input for both assays. 

Deamination reaction implemented in the OCA-Plus assay was conducted using Uracil-DNA Glycosylase—heat labile (Thermo Fisher Scientific, Waltham, MA, USA), prior to polymerase chain reaction (PCR) amplification. 

For sequencing, prepared libraries were loaded according to manufacturer’s instructions (Ion 550™—Chef, MAN0017275 (Revision C.0)) onto Ion 550™ Chips (Thermo Fisher Scientific, Waltham, MA, USA) and prepared using the Ion Chef™ System. Sequencing was performed using the Ion S5™ XL Sequencer (Thermo Fisher Scientific, Waltham, MA, USA). The data was mapped to the human genome assembly 19, embedded as the standard reference genome in the Ion Reporter™ Software (v. 5.14) (Thermo Fisher Scientific, Waltham, MA, USA). We used the Ion Reporter™ Software for initial automated analysis. Oncomine Comprehensive Plus—w2.0—DNA—Single Sample was used as analysis workflow for OCA-Plus samples and Oncomine Comprehensive v3—w4.0—DNA—Single Sample as analysis workflow for OCAv3 samples. Additionally, coverage analysis reports from the Ion Reporter™ Software providing measurements of mapped reads, mean depth, uniformity and alignment over a target region were used as quality assessment of the sequencing reactions.

### 2.4. Data Analysis

Files for analysis were provided from the Ion Reporter™ Software (v. 5.14), with use of data interchange standard parameters defined in via a JavaScript Object Notations file. Files were downloaded without any filter chain, providing all identified variants. All data analyses were conducted using Python programming language (v.3.7). 

#### 2.4.1. Gene and Nucleotide-Position Filtering

Latest versions of Browser Extensible Data (BED) files for OCA-Plus (file version: 20191203) and OCAv3 (file version: 20180509) available on the Ion Reporter™ Software (v. 5.14). Intersecting genes were identified using gene-id from both BED files and subsequently used as a filter only to filter variants within the list of intersecting genes. A list of intersecting genes and genes unique to OCAv3 and OCA-Plus is outlined in Appendix A. From the BED file of OCAv3, the assay was found to cover 146 genes with the requirement of DNA for analysis. The OCAv3 is set to cover 161 genes; nevertheless, the fusion genes *ERG*, *ETV1*, *ETV4*, *FGR*, *JAK2*, *MYB*, *MYBL1*, *NOTCH4*, *NRG1*, *NUTM1*, *PRKACA*, *PRKACB*, *RELA*, *RSPO2* and *RSPO3* were not included in the BED file of OCAv3. We have subjected DNA for this comparative analysis. Therefore, the analysis of the latter mentioned fusion genes were not examined nor included, thus requiring RNA as input and the corresponding BED file for the fusion genes. Specific covered locus positions were obtained from BED files using the provided amplicon description of start and end position. Every position between start and end was extracted, and overlapping locus positions were applied as an additional filter to assure that genes and additional locus position could be rightfully compared during analyses.

#### 2.4.2. Pre-Analysis Data Cleaning

Data was initially pre-filtered, considering variants within exonic regions or splice site regions (variants located within the first 3 nucleotides of the 5′ or 3′ end), classified as either SNV, MNV or indel, as follows: Location = “*exonic*” or “*splicesite*”;Ion Reporter™ filter = “*PASS*”;Nucleotide length ≥ 1;Variant type = *SNV*, *MNV* or *indel.*

#### 2.4.3. Original Variant Filtering

Variants passing the Pre-analysis data cleaning (2.4.2) were subsequently analyzed and annotated based on the variant filtering properties outlined below. Briefly, coverage and Phred score were based on cut-offs employed in our current clinical setting. Moreover, the Ion Torrent platform has a known limitation in homopolymeric regions, exhibiting lower accuracy when reading lengths greater than 5 bp of the homo-nucleotides [19,20]. The filtering was performed, and each variant were flagged as described below: Synonymous variant: The Ion Reporter™ Variant Effect = “synonymous”;Common SNP: UCSC Common SNP = “Common SNP”;Above *p*-value: Ion Reporter™ *p*-value > 0.01;Low overall coverage: coverage < 100;Variant with allelic ratio below Q1: allele ratio below 25% of mean allele ratio per sample;Potential germline: allele ratio on target allele = 1;High homopolymer content: Homopolymer length ≥ 5;Low base coverage: coverage < 10% of mean coverage above 100;Low Phred score: Phred score < 200;PASS: variants that passed all above criteria.

#### 2.4.4. Benign and Germline Variant Filtering

Considering that samples were not normal- and tumor-matched, we inferred potential germline variants based on “1000 Genomes” and “GnomAD/ExAc” databases [21]. Variants were further cross-referenced and clinically annotated using the database provided in Varsome using the implemented ClinVar database verdict. Variants classified as benign or likely benign germline and reviewed by expert panels on ClinVar were excluded from the variants identified. Excluded variants are outlined in Table 2. 

#### 2.4.5. Rescue Variant Cleaning and Filtering

Variants isolated for rescue included variants with an allele ratio below Q1 or variants with low base coverage identified from Section 2.4.3. Moreover, variants inadequate for NGS analysis annotated as *Nocall* by the Ion Reporter™ were also subjected for filtering. For instance, a *Nocall* variant could be inadequate for analysis due to low number of reads (reads < 25). The purpose of filtering these groups is to explore if true variants reside within this group, due to possible assay specificities not being optimized. These variants were cleaned for variants being common SNPs, synonymous mutations and variants absent of Ion Reporter™ *p*-value. 

Below Q1 variant filter: Allele ratio ≥ 11%;Ion Reporter™ *p*-value ≤ 0.001;Phred score ≥ 200;Coverage ≥ 200.

Low base coverage-variant filter: Allele ratio ≥ 11%;Ion Reporter™ *p*-value ≤ 0.001;Phred score ≥ 200;Coverage ≥ 70.

*Nocall* variant filter: Allele ratio ≥ 30%;Ion Reporter™ *p*-value ≤ 0.001;Phred score ≥ 200.

### 2.5. Statistical Analysis

All statistical analyses were performed by Python programming language (v. 3.7) using the SciPy (v. 1.6.3) package.

Fisher’s Exact test was used to calculate and determine strand bias. 

Chi-square test was applied for determining statistical differences in the number of base substitutions present between OCAv3 and OCA-Plus. 

Shapiro–Wilk test was used to determine if the coverage of *PTEN* across all 100 sequencing samples (50 × OCAv3 and 50 × OCA-plus) was normally distributed [22]. Normal distribution was rejected if the *p*-value was below 0.05. 

Wilcoxon-rank-sum test was applied to determine if *PTEN* exhibited significant statistical lower coverage in OCA-Plus contra in OCAv3. 

Pearson correlation coefficient was used to determine the linear relationship between allele ratios of final identified variants. 

## 3. Results

We aimed to compare the OCAv3 and the newly OCA-Plus to report the concordance between the identified variants in 50 paired OC FFPE-tissue samples. Moreover, this comparison was performed in order to assess the viability of OCA-Plus in our current clinical setting. For the assessment, we systematically designed a workflow to filter all identified variants obtained from sequencing to comprehensively validate the assays (Appendix A). The final variant identification was also coupled with the assessment of prior treatment of DNA with UDG, a feature incorporated in the workflow of OCA-Plus. In that manner, we were able to evaluate background noise from low-frequency variants possibly originating from deamination. 

Initially, the exploratory data analysis of unfiltered variants showed that, on average, OCA-Plus exhibited 7489 variants (±196) per sample, whereas OCAv3 displayed 3371 (±71) (Appendix A). Nonetheless, the OCAv3 design covers 146 genes using 3781 amplicons, whilst OCA-Plus covers 501 genes with 13,473 amplicons, hence the observed differences in the number of variants are expected.

Based on our routine clinical use of OCAv3, we considered valid sequencing reactions to harbor at least 5 million mapped reads, a mean depth equal to 1000 or above, a uniformity of minimum 80% and an on-target percentage above 80. As OCA-Plus contains 3.5 times more primers, we assume a valid sequencing reaction would contain at minimum 17.5 million mapped reads. The average numbers from the coverage analysis reports are shown in Table 3. In consequence of the additional 357 genes, the mean depth observed was lower for OCA-Plus compared to OCAv3 (Table 3). 

### 3.1. Filtering on Variants with Overlapping Genes and Nucleotide Positions

To initially start the comparison of variants identified in the 50 sample pairs of OCAv3 and OCA-Plus, we set out with a total of 542,753 identified variants (Appendix A), with 69% (374,230/542,753) identified from OCA-Plus and the remaining 31% (168,523/542,753) being from OCAv3. To perform the comparison between them, we only considered genes and residue positions covered in both assays (Figure 1). 

Filtering variants on intersecting genes left a total of 343,286 variants (Appendix A). Subsequently, filtering on overlapping nucleotide positions determined a normalized starting point for further filtering and assessment of OCA-Plus. Variants of OCA-Plus accounted for 50.7% (162,456/320,397) and variants of OCAv3 accounted for 49.3% (157,941/320,397) of total variants after filtering (Appendix A). 

### 3.2. Data Cleaning of Variants

To proceed filtering of variants (*n* = 320,397), we devised a pre-analysis data cleaning filter (Appendix A) designed to remove variants that are not located within an exonic region or within a splice site (three nucleotides before or after an exon). Moreover, variants that were not annotated as *PASS* from the Ion Reporter™ filter were also removed. The *PASS* annotation is indicative of adequate quality of the variant to be subjected for analysis with standard variant finding parameters provided in the Ion Reporter™ Software. Lastly, copy number variants were excluded from the analysis, as these require special filtering and analysis scripts/pipelines. Thus, variants classified as SNV, MNV or indels were considered, leaving a total of 10,836 variants for further analysis (Appendix A). 

### 3.3. Original Filtering of Variants

The correct identification of variants is crucial for diagnosis, prognosis, therapeutic treatment and/or response. Therefore, variants need to be identified with a high degree of confidence. We created a filter containing validity parameters such as Phred-score, *p*-value, allele frequency and coverage. Moreover, the filter also removed variants such as common SNPs, synonymous mutations and potential germline. For the latter, variants were annotated if the allele ratio of the variant identified was equal to 1, assuming an ideal germline variant allele frequency when homozygous [23,24]. The population allele frequency defining common SNP was ≥1% in minor allele frequency. The full list of parameters is shown in Appendix A, and specific criteria are outlined in the Materials and Methods Section 2.4.3 under “Data analysis”. Post filtering showed that 223 variants were identified in OCA-Plus in contrast to 235 variants via OCAv3 (Appendix A). 

### 3.4. Benign/Germline Variants

Of the variants identified (*n* = 458), we encountered that the Ion Reporter™ Software annotates variants from the National Center of Biotechnology Information (NCBI) Clinvar databases from 2019 (latest version 20190909). Therefore, all variants were manually inspected and cross-referenced with updated mutational verdicts from VarSome and the Catalogue of Somatic Mutations in Cancer (COSMIC). All variants were further examined using the ClinVar database as some variants were not provided with a COSMIC ID due to their novelty. Manual inspection revealed that *ARID1A*, *BRCA-1*, *BRCA-2*, *PMS2*, *TP53* and *TSC2* harbored benign/likely-benign/germline gene mutations. The specific mutations found in the latter mentioned genes are outlined in Table 2, and variants were further filtered for these specific locus mutations. The final variants were reduced to 187 and 183 for OCAv3 and OCA-Plus, respectively (Appendix A). Benign/likely-benign/germline variants are not biologically druggable and hence should be prospectively classified as common SNPs. Variants not annotated as being a common SNP, when truly confirmed, may disturb interpretation of critical variants in a clinical setting. Thus, it is appropriate to cross-reference variants with updated databases.

### 3.5. Variant Rescue Filtering

After filtering, we inspected the list of variants, counting a total of 370 variants (Appendix A). Noticeably, the pathogenic variant found in *PTEN* (chr10:89692904) in one sample of OCAv3 was absent in the corresponding sample of OCA-Plus. The *PTEN* variant found in this sample of OCAv3 exhibited a coverage of 1688, an allele frequency of 93.07%, a Phred score of 22,066 and a *p*-value of 0.00001. Additionally, we found no strand bias (*p* = 0.702 (Fisher’s exact test)). Surprisingly, the exploration of the corresponding sample in OCA-Plus revealed a coverage of 21, an allele frequency of 90.48%, a Phred score of 231, a *p*-value of 0.00001 and no strand bias (*p* = 0.185 (Fisher’s exact test)). Thus, the variant was captured by our filtering due to low coverage. Furthermore, we encountered that this specific locus position—chr10:89692904—in all 50 samples of OCA-Plus showed a statistically significantly lower coverage of 15.28 (±9.13) in contrast to a uniformly high coverage of 1901.64 (±229.45) in OCAv3 (*p* < 0.00001 (Wilcoxon rank sum test)). By this observation, we were encouraged to examine and potential rescue variants that was filtered due to low base coverage or variants with an allele ratio below the first quartile. 

Hence, cytosine deamination artifacts have been reported to cause baseline noise among low-frequency variants [14,24], we investigated if there was a clear indication of deamination artifacts among variants below the first quartile. The use of the Chi-square test revealed no statistical difference in C > T (*p* = 0.54) and G > A (*p* = 0.53) between OCAv3 and OCA-Plus in variants above the first quartile (Figure 2A), nor did the remaining base substitutions show statistical differences. Despite that, variants with an allele frequency below the first quartile revealed a statistically significantly differences in C > T and G > A substitutions (Figure 2B). OCA-Plus exhibited a significant lower amount of C > T (*p* < 0.00001 (Chi-square test)) and G > A (*p* < 0.00001 (Chi-square test)) substitutions, presumably due to initial treatment of DNA with UDG (Figure 2B). Moreover, A > G (*p* < 0.00001 (Chi-square test)) and T > C (*p* < 0.00001 (Chi-square test)) transitions were also observed significantly elevated in OCA-Plus, hence, its appearances are not clear (Figure 2B).

To define an adequately lower allele ratio threshold dealing variants with C > T or G > A substitutions, we examined at which point the allele ratio fluctuated. Kernel density estimation revealed a clear peak of C > T and G > A transitions around an allele frequency of 5% (Figure 3). The density of C > T and G > A shows a higher yield in OCAv3 samples; thus, this is expected as samples were not treated with UDG. Based on these observations we defined a threshold of 11% in accepted allele ratio for variants that had an allele ratio below the first quartile. Small peaks around an allele ratio of 0.13 were observed for both assays (Figure 3), however these peaks harbored low density compared to their respective parent peaks and therefore were considered as insignificant noise for further analysis. 

From our rescuing filtering, we restored a total of 16 variants. From the group of variants annotated as variants with an allele ratio below Q1, we rescued 2% (10/501) (Appendix A) via the devised rescue filter for this group of variants. Moreover, 5 out of 111 variants were rescued from the group harboring low-base-coverage variants (Appendix A). Moreover, devised filtering to account for true variants within the group of *Nocall* restored *PTEN* (Appendix A). The rescue variants are outlined in Table 4.

### 3.6. Final Variants

A total of 386 variants were identified with 190 variants identified using OCA-Plus, and 196 variants found were identified using OCAv3 after filtering, including variants from rescue filtering, accounting for 4.4% (17/386) of the total (Appendix A). The distribution of the ClinVar database variant classification is showed in Table 5. 

Moreover, we observed that the actual count of variants varied slightly between OCA-Plus and OCAv3 (Figure 4A). On average, OCA-Plus displayed 3.80 (±2.08) variants per sample, whereas OCAv3 demonstrated 3.92 (±2.08) variants per sample. Complete overlapping variants within both assays were assessed using locus position and amino acid change. Analysis showed a complete overlap in 74% (37/50) of the sample pairs (Figure 4B), including all variant verdicts. For a quality assessment parameter to inspect if variant were identified with approximately the same allele ratio, we applied a Pearson correlation coefficient. The analysis using a Pearson correlation coefficient showed a strong positive correlation (r = 0.83, *p* < 0.00001) between the allele ratios detected for the same variants in the two assays (Appendix A). 

To further proceed evaluation of variants identified in OCA-Plus and OCAv3 we examined the 26% (13/50) that contained non-overlapping variants counting 4.4% of incidences (17/386) (Table 6). Interestingly, within these 4.4%, the variants of OCAv3 were harboring G > A or C > T transitions in 54% (6/11) of the cases. By examination of these specific locus positions in OCA-Plus corresponding samples via Integrative Genomics Viewer (IGV) [25] (v. 2.5.2) revealed no variant. Consequently, leaving these variants of OCAv3 to be classified as potential deamination artifactual variants. Strand bias was observed in 28% (2/11) both being in *NOTCH1*, leaving *PIK3R1* (p.Ser412IlefsTer6) and *CREBBP* (p.Leu551Ile) to be potential true variants, although being classified as variant of uncertain significance. However, the variant of *TP53* (p.Thr102AsnfsTer47) was also present. The inspection of this variant via IGV showed that the variant correctly identified in OCAv3 was centrally located in the amplicon covering this region. However, the amplicon covering this region in OCA-Plus have been altered, so the variant is not called although present.

OCA-Plus exhibited an attentional mutational signature of T > C transitions in 33% (2/6) of the disconcordant variants. Uncertainties of these being potential artifactual variants is present, as deamination of adenine to hypoxanthine; thus, base pairing cytosine would explain the observed after subsequently artificial replication. Although the leading cause to these variants are unknown. Hence, we speculate if initial UDG treatment can be involved. Another possibility could be technical amplification error orchestrated during early PCR. *PMS2* (p.Asp526Gly) and *CREBBP* (p.Pro2285AlafsTer56) was observed subject to strand bias (*p* < 0.0001 (Fisher exact test)), leaving 66% (4/6) as potential true variants being *BRCA1* (p.Thr1349AsnfsTer7), *NTRK1* (p.Arg780Gln), *RAD51B* (p.Met120Thr) and *BRCA2* (p.Glu1879Lys).

As benign-variants and VUS are not used for diagnostics and treatment discussion, we further emphasized on examining variants of likely pathogenic or pathogenic classification by ClinVar. From our identified variants, we encountered a *TP53* splice site variant harboring a T/G transversion at position chr17:7577610 without any provided verdict classification from ClinVar. From inspection, the variant is located at the beginning of exon 6. As the disruption of a splice sites often effectuates a truncation of the gene product, hence affecting protein function, we classified this variant as likely pathogenic. 

We observed that four sample pairs did not harbor likely pathogenic and pathogenic variants. In 91% (42/46) of samples, variants were identified in both samples (Figure 5). The 9% (4/46) varying between samples was assigned to the variants of *NF1*, *POLE* and *TP53* (Table 6 and Figure 5).

## 4. Discussion

In this study, we compared variants identified on 313,769 possible intersecting nucleotide positions within 144 genes covered in both the OCAv3 and the novel OCA-Plus panels. Moreover, the OCA-Plus provides the possibility to address additional 357 cancer associated genes and TMB and MSI analyses. Nonetheless, in general, high-grade serous adenocarcinoma cases do not present high chromosomal instability, corroborating our findings (Appendix A). Therefore, cases with known high genomic instability would be better suited for evaluating the TMB and MSI performance. Here, we used a cut-off of ≥ 20 for TMB-high, according to previous studies and our current clinical setting [26]. Moreover, the MSI-high was defined by following the recommendations from the manufacturer, with a score ≥ 26. Therefore, we did not validate the performance of these features. 

The OCA-Plus, moreover, takes advantage of the possibility of pre-treating FFPE sample DNA with UDG; thus, it is reported to decrease the amount of C:G > T:A deamination artifactual noise when interpreting NGS results [27,28]. We did confirm this observation with a reduction of 31% in OCA-Plus of C:G > T:A substitutions among low frequency variants. To begin the evaluation of OCA-Plus in contrast to OCAv3, we created a study design of 50 FFPE samples of OC patients. In comparison of the two assays of paired samples, we initially used the Ion Reporter™ Software to generate variant calling files for analysis. Using our designed filtering criterions, we found that there was a 91% concordance between pathogenic and likely pathogenic classified variants identified in the two assays. Differences were observed in the *NF1* (chr17:29553538), in two cases, *POLE* (chr12:133235881) and additionally *TP53* (chr17:7579382) of OCAv3. The concern of *TP53* arises as previous sequencing data using a *TP53-*specific gene panel identifies the *TP53* variant in sample 29 (Table 6), thus proving its validity. None of the other variants classified as pathogenic or likely pathogenic were validated by an orthogonal method. Although we speculate if the variants of *NF1* and *POLE* are artifactual variants due to their transition of C:G > T:A. The examination of these variants via IGV revealed no variant. Albeit a prominent overlap of identified variants, we want to address interesting observations from our analysis, including *PTEN* and *TP53* variants.

Moreover, the OCA-Plus is considerably more explorative by covering approximately 3.5 times more genes than covered in the OCAv3. By this substantial difference in genes covered, it is expected to encounter additional information concerning the genetic profile underlying the samples. However, this additional information was not addressed, as only overlapping gene and nucleotide positions were examined during this comparative study of OCAv3 and OCA-Plus.

### 4.1. PTEN and TP53 Amplicon Observation

The OCA-Plus is constituting 13,737 amplicons covering 501 genes related to cancer and is thus an upgrade of the OCAv3 for further mutational exploration in additional cancer genes. For a systematic comparison, we compared variants spanning 313,769 specific locus positions covered in both assays. During the analysis, we encountered that the pathogenic variant at position chr10:89692904 of *PTEN* was poorly covered with an average coverage of 15.28 (± 9.13) across all 50 samples analyzed with OCA-Plus, hence filtered from original variants. By this observation, we found it necessary to explore variants in filtered groups. We did manage to restore the variant of *PTEN* as it clearly stood out with its high allele frequency within its filtered group. We speculate that the reason for this observation is a consequence of specific primer dimerization, with the primers covering this region of *PTEN*. Thus, primers should be revised for this area for improving coverage. 

*PIK3CA* encodes the catalytic subunit of PI3K performing phosphorylation, triggering the downstream signaling of cell growth/proliferation and cell survival *inter alia*, facilitating uncontrolled cell growth when the genome is harboring pathogenic *PIK3CA* mutations [29]. The biological drug, Alpelisib, is specifically inhibiting PI3K-signaling in the PI3K/AKT signaling pathway [30]. The phosphatase PTEN is contributing the regulation of PI3K phosphorylation, hence controlling PI3K/AKT downstream signaling [31,32]. However, somatic mutation causing loss-of-function of the PTEN also primes elevated PI3K/AKT, thus accelerating cell growth/proliferation and cell survival. Patients harboring pathogenic *PTEN* mutations may therefore benefit targeted treatment with Alpelisib. The identification of variants within *PTEN* is therefore of importance for tailoring treatment.

Moreover, oncologists and molecular biologists use the tumor percentage for the interpretation of results in a clinical setting. The *TP53* gene harboring a pathogenic mutation is an often-used measurement in determining the tumor percentage. We did, however, encounter a disconcordant variant of *TP53*, absent in an OCA-Plus corresponding sample. By inspection, we could conclude that amplicon covering this area of *TP53* has been changed in the OCA-Plus. Hence, this region should be manually inspected or validated via a *TP53* gene panel for correct interpretation.

### 4.2. Unexplained Sequence Artifactual Variants

The C:G > T:A deamination of artifactual variants is a known baseline noise in DNA extracted from FFPE tissue [15]. Although variants caused by deamination are commonly observed with low allele frequency, artifactual variants can also be observed with higher allele frequency. These can arise if DNA integrity is compromised and hence the variant is introduced during in vitro replication [33]. However, we did observe remarkably lower levels of C:G > T:A transitions in samples subjected to UDG via initial treatment in OCA-Plus protocol. Nevertheless, noticeable from our analysis revealed that OCA-Plus samples harbored elevated yet unexplained A:T > G:C transitions. 

### 4.3. Cost Benefit

The OCA-Plus panel may be cost and labor efficient when compared to individual testing of OCAv3, TMB and MSI analyses. Moreover, OCA-Plus provides additional information of 357 genes. However, the beneficial aspect is considered based on a clinical condition that the analysis chosen benefits the patient. The optimization of combining OCA-Plus, MSI and TMB into one assay may ideally translate into optimization of workflows, labor savings and a reduction in reagents used in library preparations. These aspects are critically dependent on the patient flow within the clinical setting to account for sequencing reactions and sequencing chip use. As a low flow of patients can counteract the economic benefit by an increased amount of sequencing reagents used per patient. The assays used should reflect the patient need at a given date during disease.

## 5. Conclusions

Correct identification of mutations is important for obtaining a telling picture of the underlying genomic landscape of a tumor for an optimal therapeutic course. Although only a subset of gene mutations is clinical actionable, additional information in cancer-related genes may benefit to improved knowledge of cancer. However, gene panels must be concise not to compromise quality and read depth of variants. Moreover, choice of gene-panel should be carefully weighed to which question that needs an answer. Hence, looking for mutations only within HRR genes in a clinical setting, an HRR gene panel would be preferred. This aspect is also to avoid getting information that cannot benefit the patient and to favor more analyses that can be setup and run during the same sequencing.

In conclusion, we found that OCA-Plus can substitute OCAv3 without compromising performance. Although, not addressed in this study, the OCA-Plus may potentially provide additional information of 357 cancer-related genes, TMB and MSI. Notably, we did observe limitations due to different amplicons underlying the OCA-Plus in regions of *TP53* and *PTEN*. Thus, the manual inspection of these regions through the IGV is recommended. 

## Figures and Tables

**Figure 1 cancers-13-05230-f001:**
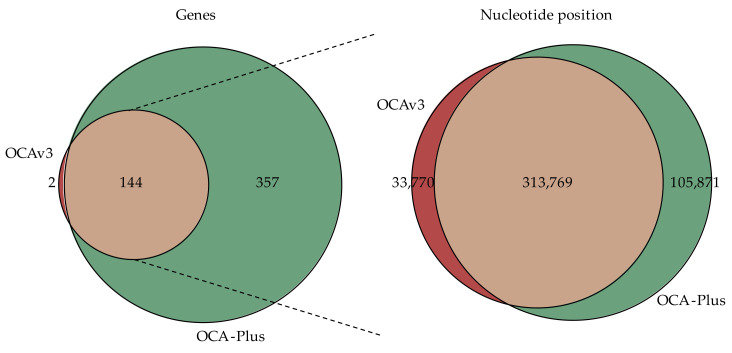
Identification of overlapping genes and overlapping locus positions. Genes and amplicon coverage of nucleotide positions were extracted from Oncomine™ Comprehensive Assay v3 and Oncomine™ Comprehensive Assay Plus browser extensible data files, respectively. From left to right: Venn diagram showing that 144 genes are intersecting between OCAv3 and OCA-Plus. Out of the 144 genes, 313,769 nucleotide positions are covered in both OCAv3 and OCA-Plus.

**Figure 2 cancers-13-05230-f002:**
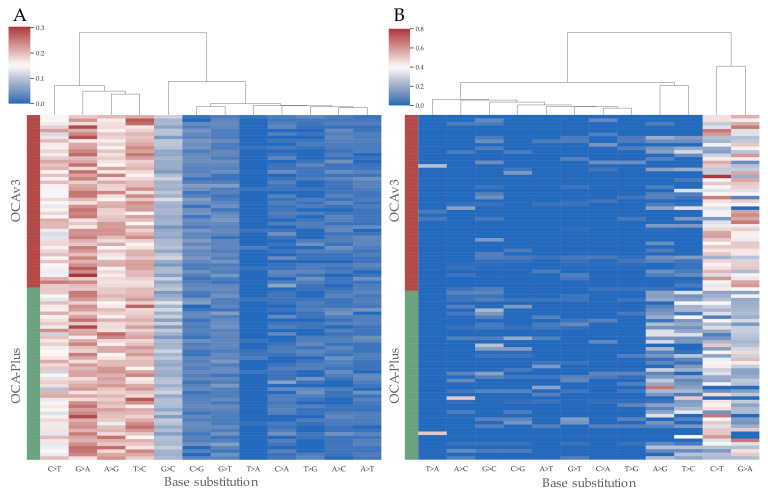
Potential sequence artifactual mutations are harbored in variants below the first quartile of mean sample allele ratios. (**A**) Cluster map of proportional base substitutions in variants above the first quartile. (**B**) Cluster map of proportional base substitutions in variants below first quartile of allele ratio. OCA-Plus harbored statistical lower levels of C > T and G > A but also significantly higher levels of A > G and T > C substitutions.

**Figure 3 cancers-13-05230-f003:**
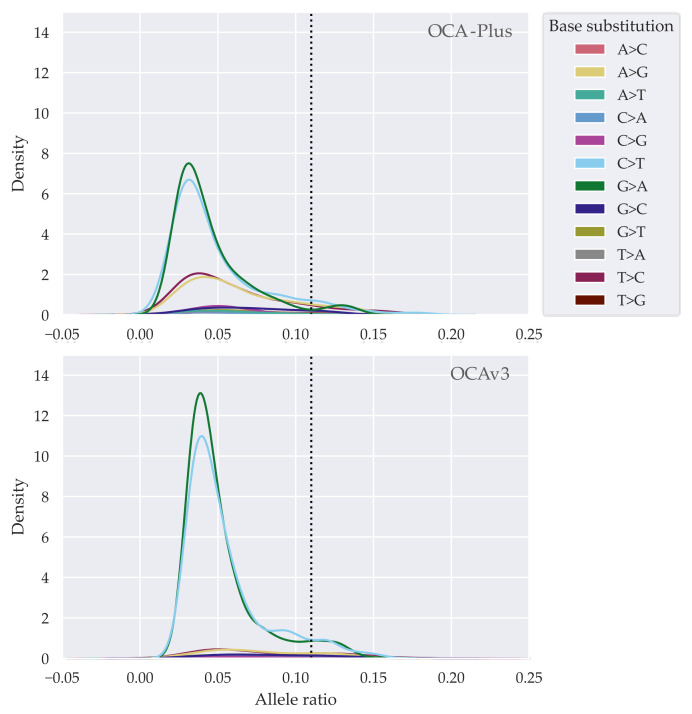
Allele frequency-based distribution of base substitutions of variants below first quartile. Dotted lines represent our determined cut-off value by an allele frequency of 11%.

**Figure 4 cancers-13-05230-f004:**
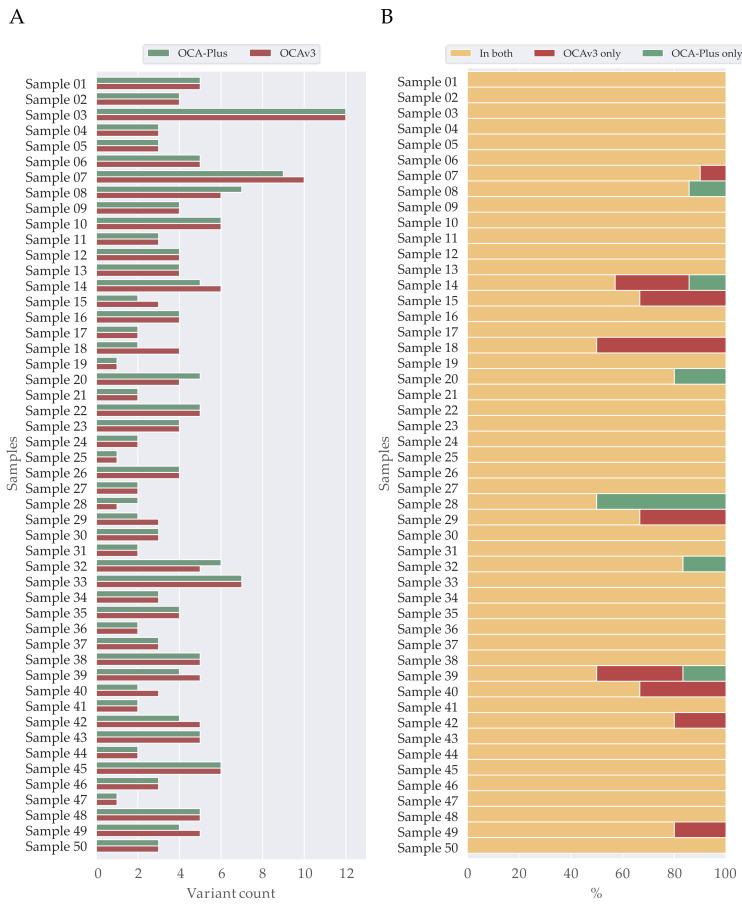
Counts and overlap of identified variants in OCAv3 and OCA-Plus. (**A**) Bar-chart showing locus variants count per samples. (**B**) Stacked bar-chart of percentual overlapping locus variants of OCAv3 and OCA-Plus per samples; 37/50 samples overlapped completely and 13/50 showed difference in locus variants, hence only identified in one of the assays.

**Figure 5 cancers-13-05230-f005:**
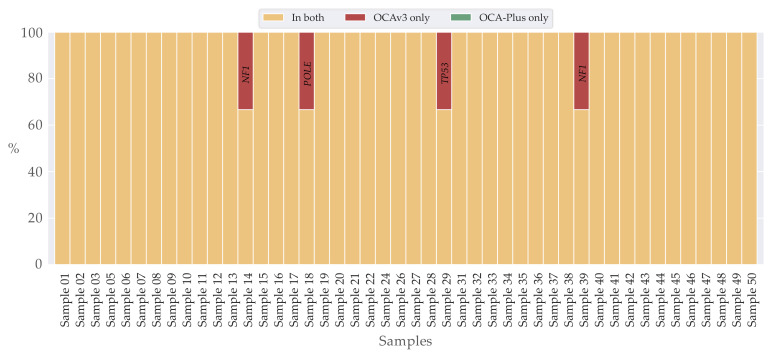
Overlap of identified variants in OCAv3 and OCA-Plus of clinical relevance. Stacked bar-chart of percentual overlapping locus variants of OCAv3 and OCA-Plus per samples; 42/46 samples overlapped completely, 4/46 showed difference in locus variants, hence only identified in OCAv3.

**Table 1 cancers-13-05230-t001:** List of selected FDA-approved molecular drugs for treatment of ovarian cancer and breast cancer patients.

Drug	Mechanism of Action	Gene	Alterations	Cancer Type(s)
Niraparib	PARP-inhibitor	*BRCA1*, *BRCA2*	Pathogenic mutations	Ovarian cancer
Olaparib	PARP-inhibitor	*BRCA1*, *BRCA2*	Pathogenic mutations	Ovarian cancer, breast cancer
Olaparib + Bevacizumab	PARP-inhibitor + VEGF receptor-inhibitor	*BRCA1*, *BRCA2*	Pathogenic mutations	Ovarian cancer
Rucaparib	PARP-inhibitor	*BRCA1*, *BRCA2*	Pathogenic mutations	Ovarian cancer
Talazoparib	PARP-inhibitor	*BRCA1*, *BRCA2*	Pathogenic mutations	Breast cancer
Alpelisib + Fulvestrant	PI3K-inhibitor + Estrogen receptor-inhibitor	*PIK3CA*	Pathogenic mutations	Breast cancer
Ado-Trastuzumab Emtansine	HER2-inhibitor	*ERBB2*	C420R, E542K, E545A, E545D, E545G, E545K, H1047L, H1047R, H1047Y, Q546E, Q546R	Breast cancer
Entrectinib	TRK-inhibitor	*NTRK1*, *NTRK2*, *NTRK3*	Fusions	All solid tumors
Larotrectinib	TRK-inhibitor	*NTRK1*, *NTRK2*, *NTRK3*	Fusions	All solid tumors
Pembrolizumab	PD-1-inhibitor	N/A	MSI-high	All solid tumors
Pembrolizumab	PD-1-inhibitor	N/A	TMB-high	All solid tumors

Abbreviations. PARP: Poly (ADP-ribose) polymerase; VEGF: Vascular endothelial growth factor; PI3K: Phosphoinositide-3-kinase; TRK: Tropomyosin receptor kinase; PD-1: Programmed cell death protein-1; MSI: Micro satellite instability; TMB: Tumor mutational burden.

**Table 2 cancers-13-05230-t002:** A list of encountered benign, likely-benign germline variants annotated from Varsome using ClinVar verdicts ^1^.

Gene	Locus	Amino Acid Change	Last Evaluation
*ARID1A*	chr1:27100181	p.Gln1334del	29 January 2020
*BRCA1*	chr17:41245027	p.Arg841Trp	1 May 2021
*BRCA1*	chr17:41244130	p.Ser1140Gly	1 May 2021
*BRCA2*	chr13:32929387	p.Val2466Ala	18 April 2021
*PMS2*	chr7:6045634	p.Ile18Val	1 May 2021
*TP53*	chr17:7577577	p.Asn235Ser	1 May 2021
*TSC2*	chr16:2134508	p.Ala1429Ser	6 April 2021
*TSC2*	chr16:2103392	p.Glu92Val	6 April 2021

^1^ Database searched were conducted 9 August 2021.

**Table 3 cancers-13-05230-t003:** Quality measurements of sequencing reactions.

Assay	Mapped Reads (mill.)	Mean Depth	Uniformity ^1^	On Target ^2^
OCAv3	11.41 (±4.44)	3100.16 (±1194.72)	85.96% (±4.11)	96.22% (±1.15)
OCA-Plus	21.45 (±4.21)	1604.94 (±328.44)	92.53% (±4.40)	93.20% (±2.11)

Dispersity of the measurements in relation to the mean are indicated by the standard deviation within brackets. ^1^ Uniformity: The percentage of bases covered by at least 20% of mean depth. ^2^ On Target: The percentage of reads aligned over a target region.

**Table 4 cancers-13-05230-t004:** Rescued variants.

Sample	Locus	Gene	Geno-Type	AA-Change	Raw Coverage	Phred Score	*p*-Value	Allele Ratio	SB ^1^	ClinVar ^2^	Filter ^3^
OCA-Plus											
Sample 03	chr10:89692904	*PTEN*	C/G	p.Arg130Gly	21	231.27	<0.0001	0.904	0.185	P	NC
Sample 04	chr11:108150246	*ATM*	TC/T	p.Arg1106GlyfsTer3	562	403.46	<0.0001	0.127	0.705	VUS	BQ1
Sample 14	chr3:142274770	*ATR*	T/C	p.Lys764Glu	94	722.92	<0.0001	0.617	0.089	B/LB	LBC
Sample 21	chr1:120491728	*NOTCH2*	A/C	p.Leu834Trp	1107	840.32	<0.0001	0.130	0.090	VUS	BQ1
Sample 32	chr8:90955552	*NBN*	C/A	p.Gly705Ter	2488	1202.65	<0.0001	0.113	0.776	P	BQ1
Sample 33	chr3:142231275	*ATR*	T/A	p.Asp1560Val	562	556.65	<0.0001	0.153	0.547	VUS	BQ1
Sample 33	chr7:6031639	*PMS2*	T/C	p.Tyr318Cys	72	267.25	<0.0001	0.388	1	VUS	LBC
OCAv3											
Sample 03	chr3:178917490	*PIK3CA*	G/A	p.Gly122Asp	70	429.41	<0.0001	0.571	0.805	VUS	LBC
Sample 03	chr3:178936091	*PIK3CA*	G/A	p.Glu545Lys	87	406.07	<0.0001	0.505	0.131	P	LBC
Sample 04	chr11:108150246	*ATM*	TC/T	p.Arg1106GlyfsTer3	738	504.82	<0.0001	0.145	0.920	VUS	BQ1
Sample 21	chr1:120491728	*NOTCH2*	A/C	p.Leu834Trp	4314	1082.00	<0.0001	0.134	0.891	VUS	BQ1
Sample 27	chr11:125514063	*CHEK1*	A/G	p.Tyr334Cys	78	313.13	<0.0001	0.423	0.443	VUS	LBC
Sample 32	chr8:90955552	*NBN*	C/A	p.Gly705Ter	3262	1383.17	<0.0001	0.152	0.852	P	BQ1
Sample 33	chr3:142231275	*ATR*	T/A	p.Asp1560Val	14,789	1232.01	<0.0001	0.143	0.521	VUS	BQ1
Sample 39	chr16:3831230	*CREBBP*	G/T	p.Leu551Ile	431	213.63	<0.0001	0.125	0.884	VUS	BQ1
Sample 46	chr17:7577518	*TP53*	T/A	p.Ile255Phe	833	460.46	<0.0001	0.134	0.175	LP	BQ1

^1^ SB: Strand bias—column showing the *p*-value of Fishers exact test. ^2^ ClinVar: Column is showing the verdict from the ClinVar database. B/LB: Benign/likely benign; LP: Likely pathogenic; P: Pathogenic; VUS: Variant of uncertain significance. ^3^ Filter: States within which filter the variant is identified. BQ1: Below first quartile; LBC: Low base coverage; NC: Nocall.

**Table 5 cancers-13-05230-t005:** Proportion of classified variants.

ClinVar Classification	Proportion
Pathogenic	26.9% (104/386)
Pathogenic/Likely pathogenic	0.5% (2/386)
Likely pathogenic	13.0% (50/386)
VUS	37.6% (145/386)
Benign	2.1% (8/386)
Benign/Likely benign	18,9% (73/386)
Likely benign	1.0% (4/386)

**Table 6 cancers-13-05230-t006:** Disconcordant variants.

Sample	Locus	Gene	Geno-Type	AA-Change	Raw Coverage	Phred Score	*p*-Value	Allele Ratio	SB 1	ClinVar 2	Filter 3
OCA-Plus											
Sample 08	chr7:6026819	*PMS2*	T/C	p.Asp526Gly	157	1905.19	<0.0001	0.835	<0.0001	VUS	Original
Sample 14	chr17:41243502	*BRCA1*	GT/.	p.Thr1349AsnfsTer7	1088	17,992.20	<0.0001	0.975	1	VUS	Original
Sample 20	chr1:156851382	*NTRK1*	G/A	p.Arg780Gln	592	1695.44	<0.0001	0.298	0.23	B/LB	Original
Sample 28	chr14:68331763	*RAD51B*	T/C	p.Met120Thr	1099	10,246.80	<0.0001	0.665	0.24	VUS	Original
Sample 32	chr13:32914127	*BRCA2*	G/A	p.Glu1879Lys	900	2434.67	<0.0001	0.290	0.50	VUS	Original
Sample 39	chr16:3778195	*CREBBP*	GGG/GCGG	p.Pro2285AlafsTer56	680	8367.50	<0.0001	0.865	<0.0001	VUS	Original
OCAv3											
Sample 07	chr5:67589243	*PIK3R1*	G/GA	p.Ser412IlefsTer6	312	675.41	<0.0001	0.298	0.06	VUS	Original
Sample 14	chr9:22006119	*CDKN2B*	G/A	p.Thr95Met	275	350.76	<0.0001	0.210	0.46	VUS	Original
Sample 14	chr17:29553538	*NF1*	G/A	p.Trp696Ter	714	244.60	<0.0001	0.104	0.06	P	Original
Sample 15	chr1:120539937	*NOTCH2*	G/A	p.Thr145Met	2707	8806.30	<0.0001	0.457	0.61	B/LB	Original
Sample 18	chr9:139390917	*NOTCH1*	G/T	p.Ala2425Asp	1142	9317.66	<0.0001	0.665	<0.0001	B/LB	Original
Sample 18	chr12:133235881	*POLE*	C/T	p.Arg1092Lys	425	253.87	<0.0001	0.138	0.48	LP	Original
Sample 29	chr17:7579382	*TP53*	G/GT	p.Thr102AsnfsTer47	1190	9235.79	<0.0001	0.703	0.79	LP	Original
Sample 39	chr17:29553538	*NF1*	G/A	p.Trp696Ter	2250	19,203.40	<0.0001	0.749	0.32	P	Original
Sample 40	chr13:32954213	*BRCA2*	C/T	p.Pro3063Ser	797	5497.47	<0.0001	0.608	0.77	VUS	Original
Sample 42	chr9:139411783	*NOTCH1*	C/A	p.Cys499Phe	599	364.22	<0.0001	0.140	<0.01	VUS	Original
Sample 49	chr16:3831230	*CREBBP*	G/T	p.Leu551Ile	2509	15,058.40	<0.0001	0.645	0.25	VUS	Original

^1^ SB: Strand bias—column showing the *p*-value of Fishers exact test; ^2^ ClinVar: Column is showing the verdict from the ClinVar database; VUS: Variant of uncertain significance; B/LB: Benign/likely benign; LP: Likely pathogenic; P: Pathogenic; ^3^ Filter: States within which filter the variant is identified.

## Data Availability

Due to sensitive information, data is available upon reasonable request.

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
