# Peer review of "Oncomine™ Comprehensive Assay v3 vs. Oncomine™ Comprehensive Assay Plus"

_cancers, 2021, doi:10.3390/cancers13205230_

Round 1

Reviewer 1 Report

In this study by Verstergaard et al, the authors report a very detailed and profound evaluation of the oncomine panels, OCAv3 and OCA-Plus. The smaller OCAv3 panel is quite commonly used across many cancer types and thus the evaluation of the larger panel in comparison, is certainly relevant for many scientists working in cancer diagnostics.

The Manuscript is very well written with very informative tables and figures and an exemplary method section. The conclusions are sound and adequate, and the reporting of the discordant genes is certainly of clinical relevance and thus very important.

Some of the discussions are related to the use of UNG treatment in the OCA-Plus assay and not the OCAv3 assay which might have also led to some of the observed differences. It is however important to note that there are no reasons which prevent the use of UNG digestion in the OCAv3 assay, too, as the enzymatic step is not interfering with the downstream protocol. While I assume the authors did not wanted to alter their standard pipeline for the comparison (and that for good reasons), I still think it could be an important comment to add for readers which wonder about UNG digestion in the OCAv3 panel.

Another important limitation is the restraint on the OCAv3 and the OCA-Plus panel for the comparison. This only allowed to compare the genes which are covered by both panels but did not allow additional performance evaluation with the additional genes in the OCA-Plus panel or its performance for TMB evaluation as that would require the use of a third method. Nevertheless, the authors clearly highlight in the discussion that their results only allow the conclusion of replacing the OCAv3 with the OCA-Plus panel without compromising performance and thus this limitation should be sufficiently discussed in the paper.

Therefore, my only remaining comment for the authors would be that it would certainly be beneficial if the authors could share their analysis pipeline on a public repository like Github as despite the good method section it would have been easier like this to follow the precise filtering steps. In addition, this would be critical for interested readers who would like to replicate the results of the authors and thus I would strongly recommend making the deposition of the code used in this manuscript obligatory for its publication.

Besides, this is a clearly well written manuscript with an appropriate experimental design that deserves publication.

Reviewer 2 Report

The paper compares 2 commercial next-generation sequencing kits where one is an updated version with an expanded number of genes. The paper would greatly benefit from a more concise presentation of the data and an English grammar check. The important messages are somewhat obscured by the wealth of detail provided for each analysis.

  1. The data presented is quite detailed and could be substantially simplified. For example, Figure 2 adequately describes the overlapping genes and locus positions. The legend text could essentially replace most of the text of section 3.1 for a simple and concise summary and these details could be placed in a supplement.
  2. A major message for readers is whether the OCA-plus method identified the same variants as the OCAv3 that were considered to be pathogenic or likely pathogenic, after applying the appropriate filters, rather than full detail of the filtering. I suggest that this is the information that is discussed first in the results section. It is standard practice to remove common germline variants and it is therefore not necessary to provide the detail of how and why this was done. A simple few sentences should suffice for the filters that were applied.
  3. Was the Catalogue of Somatic Mutations in Cancer (COSMIC) database examined when considering pathogenicity of the variants? This is a highly relevant database when considering cancer specimens.
  4. Did the OCA-plus method detect additional pathogenic variants in genes that were not present in OCAv3. This is a critical consideration when discussing the advantages of one kit over another.
  5. Were any of the pathogenic or likely pathogenic variants validated by an orthogonal method? This is particularly relevant for the variants that were not detected by both methods to establish the reliability of detection of true variants.
  6. The OCA-plus method has additional benefits such as MSI and TMB. Were these features assessed for the 50 samples? Again, this is a critical consideration when discussing the advantages of one kit over another.
  7. Is there a substantial cost difference between the kits. It is important to detail the additional benefits of using the OCA-Plus kit, including the potential additional relevant variants due to the increased number of genes, the ability to identify MSI and to calculate TMB.

Minor points

  1. In the introduction ‘OC harbors a heterogenetic molecular genotype’. Should this be heterogeneous rather than heterogenetic?
  2. There is repetition in the following in the Introduction and the abbreviations should be given at first use:

The Oncomine™ Comprehensive assay Plus (OCA-Plus) covers 501 cancer-associated genes, of these 144 overlapping with OCAv3, and includes MSI and TMB assays in the same run, providing a time-efficient single workflow and faster transition from sample to results. Furthermore, OCA-Plus includes assays for microsatellite instability (MSI) and tumor mutational burden (TMB)

  1. ‘A total of 50 tissue samples were collected from a prospective cohort of OC patients [15,16].’ It is not clear why the 2 papers from 2017 are referenced when the current study was performed on a prospective cohort. Please clarify, was the NGS performed on a retrospective cohort?
  2. What was the samples input DNA concentration.
  3. What was the genome build that was used to map the data.
  4. Please clarify why the fusions genes of OCAv3 were not included in the BED-file?
  5. Please define the cut-off population allele frequency that was used to infer variants as potential germline.
  6. The term germline mutations should be replaced with germline variants.
  7. Figure 1 contains a lot of detail related to the filtering and would be more appropriate in the supplement.
  8. Table 3 – please explain what is the data in brackets in the Mean depth, Uniformity and On target columns. I assume it is standard deviation?

Round 2

Reviewer 2 Report

The authors have not addressed major concerns

The simple summary states

In this study, we compared and assessed the performance of the newly Oncomine™ Comprehensive Assay Plus including 501 cancer-related genes, microsatellite instability and tumor mutational burden assays.

There was no assessment of the microsatellite instability and tumor mutational burden assays. Therefore, this statement appears to be misleading.

The abstract states that the ‘OCA-Plus can provide a more in-depth mutational profile of genomic variants compared with OCAv3, without compromising performance.’

The authors have made no attempt to assess the performance of the new genes or features not already included in OCAv3. Therefore, this appears to be an inaccurate statement. The authors have only compared the same genes already included in OCAv3 and ideally this should be made very clear in the summary and abstract.  

The authors must first demonstrate that the original OCAv3 method in their hands generates reliable and reproducible results, which was why the question was asked - Were any of the pathogenic or likely pathogenic variants validated by an orthogonal method? This is particularly relevant for the variants that were not detected by both methods to establish the reliability of detection of true variants.

The authors did not answer this question adequately.

qPCR is mentioned, which one assumes is quantitative PCR? qPCR is used for gene expression assessment, not for variant validation.

The authors could add a simple statement that none of the variants categorized as pathogenic or likely pathogenic were independently validated using an orthogonal method.

The answer to the question related to the cost benefit. Section 4.3

The OCA-Plus panel in cost and time more effective when comparing it to OCAv3-, TMB-, MSI- and Loss-of-heterozygosity (LOH)-analysis combined.

I cannot understand this sentence. Are the authors stating that it is more cost effective? The sentence structure appears to be incorrect.

Round 3

Reviewer 2 Report

The authors have adequately addressed my concerns